# On Plasticity, Invariance, and Mutually Frozen Weights in Sequential Task Learning

**Julian Zilly**
ETH Zürich
jzilly@ethz.ch

**Alessandro Achille**
Caltech
aachille@caltech.edu

**Andrea Censi**
ETH Zürich
acensi@ethz.ch

**Emilio Frazzoli**
ETH Zürich
efrazzoli@ethz.ch

## Abstract

Plastic neural networks have the ability to adapt to new tasks. However, in a continual learning setting, the configuration of parameters learned in previous tasks can severely reduce the adaptability to future tasks. In particular, we show that, when using weight decay, weights in successive layers of a deep network may become "mutually frozen". This has a double effect: on the one hand, it makes the network updates more invariant to nuisance factors, providing a useful bias for future tasks. On the other hand, it can prevent the network from learning new tasks that require significantly different features. In this context, we find that the local input sensitivity of a *deep model* is correlated with its ability to adapt, thus leading to an intriguing trade-off between adaptability and invariance when training a deep model more than once. We then show that a simple intervention that "resets" the mutually frozen connections can improve transfer learning on a variety of visual classification tasks. The efficacy of "resetting" itself depends on the size of the target dataset and the difference of the pre-training and target domains, allowing us to achieve state-of-the-art results on some datasets.

## 1 Introduction

Lifelong learning [1, 2] holds both the promise of benefitting from past experiences and the challenge of having to continually adapt to new problem settings. The characteristics of this intriguing sequential learning problem make it a balancing act between retaining knowledge and adapting to new experiences referred to as the *stability-plasticity dilemma* [3]. Similar to a coach or teacher, we want to understand when a sequence of tasks helps or hinders further learning. Clearly, some pre-training tasks are highly beneficial [4–7] while others can hurt performance [8, 9].

The existing literature on pre-training and curriculum learning sheds some light on the desired characteristics of sequential learning. In some cases, following a sequence of learning tasks can lead to better results than simply training on the target task from scratch [10]. Work by Achille and Soatto [8] contrasts this picture by highlighting that neural networks can suffer from pre-training on some tasks. In an extreme case, when learning data is corrupted for a sufficiently long period, even switching back to clean data does not allow the neural network to recover its original performance.

*Contribution:* We formalize the basic ingredients of *sequential task learning*, introduce *mutually frozen weights* and establish a connection between plasticity and invariance of deep networks. To understand *mutually frozen weights*, we investigate their weight update dynamics, show that such frozen weights are different from regular sparse weights that are not connected to other sparse weights and that frozen weights can lead to a decrease in performance when retraining on a new task. Based on this analysis, we show that applying an *intervention* which "resets" frozen weights can improve retraining performance on a number of tasks as long as sufficiently many samples are available for retraining and even achieve state-of-the-art results on FashionMNIST image classification.

35th Conference on Neural Information Processing Systems (NeurIPS 2021).

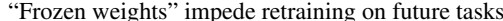

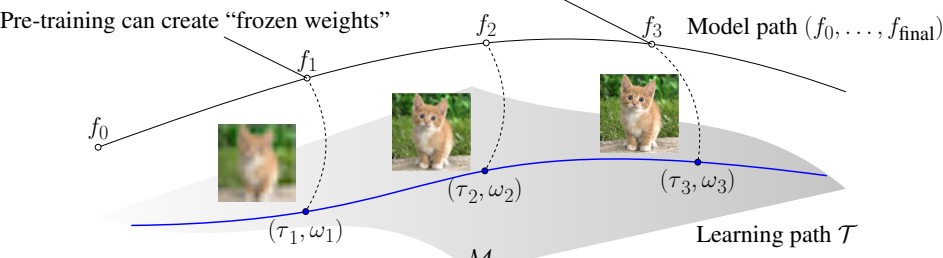

"Frozen weights" impede retraining on future tasks

Pre-training can create "frozen weights"

$f_2$  $f_3$  Model path $(f_0, \ldots, f_{\text{final}})$

$f_1$

$f_0$

$(\tau_1, \omega_1)$  $(\tau_2, \omega_2)$  $(\tau_3, \omega_3)$

$\mathcal{M}$  Learning path $\mathcal{T}$

Figure 1: In the context of the *negative pre-training effect*, e.g. when pre-training on blurred images before training on unblurred images [8], we explore the effect of "frozen weights", which can hinder retraining in a sequential learning setting, on the *generalization performance* of neural networks . A point on the learning manifold $\mathcal{M}$ is a supervised learning task $\tau := (p(x, y), \mathcal{L})$, composed of a data distribution of input and labels and a loss function, and a learning process $\omega$ which determines how a model $f$ is adapted given a task $\tau$. The learning path $\mathcal{T}$ determines model path changes from initial to final trained model $f_{\text{final}}$. We focus on the effect of *frozen weights* and the *plasticity-invariance trade-off* that emerges when sequentially training a *deep* network with weight decay.

## 2   Related work

We call the problem setting in this work *optimal task tracking* which involves training a model on a sequence of tasks to improve performance on a final target task. This strongly resembles the objectives of lifelong learning [1], continual learning [11] and meta-learning [12–14] and connects to many time-dependent learning aspects in relation to neural networks.

**Time-dependent learning:** We are aware of several time-dependent phenomena in training neural networks such as *catastrophic forgetting* [15], critical learning periods [8], and time-sensitivity of regularization and data augmentation [16]. Achille et al. [8] show that by dropping out certain frequencies during the beginning of training, networks are unable to recover to full performance, even when those frequencies are reintroduced. Similarly, Liu et al. [17] show that first optimizing on random labels ruins performance on clean data shown indefinitely to the same network afterwards. Recent information-theoretic analysis [18] suggests that neural network have distinct learning phases: First a phase where network parameters grow in information content, before, in a second phase, self-regularizing and pruning away irrelevant information. More commonly known time-dependent or sequential heuristics are learning rate schedules [19] which may even be cyclical [20], regularization annealing such as dropout annealing [21], student-teacher model transfer [22], and pre-training [23]. Finally, our introduced concept of *mutually frozen weights* also touches upon the *lottery ticket hypothesis* [24] and may help explain some of its underlying gradient dynamics.

**Curriculum learning and generation:** Curriculum learning [25, 10] is the technique of training agents by showing them a progression of tasks from easier variants to harder examples to attempt to solve the final target task. Both the utility of transferring between similar or dissimilar tasks is debated, with some results pointing towards similarity [26–28] and others towards dissimilarity [29, 30]. Meta-learning [13, 31] is similarly related by focusing on speeding up or enhancing performance on a target task by pre-training on other tasks to first *learn to learn*. Unlike curriculum learning, curriculum generation seeks to find a path through task space, generating an agent that can solve some final, held-out task [32–36].

**Invariance-plasticity-generalization:** Empirically, Mehta et al. [37] showed that ReLU networks with batch normalization trained with gradient descent lead to filter-level sparsity. Evci et al. [38] demonstrate that training sparse networks usually leads to worse outcomes than without sparsity. On the other hand, Bartoldson et al. [39] showed that increased sparsity through pruning can lead to better generalization. Bringing sparsity emergence and difficulty of training together, we will demonstrate how *certain types* of sparsity can lead to problems in retraining a model after pre-training in the context of *mutually frozen weights*. From a theoretical viewpoint, Neyshabur et al. [40, 41] connect the *path norm* of a network to its generalization via Rademacher bounds, where the *path norm* is based on the sum across all paths of the product of weights associated to individual paths.

# 3 Optimal task tracking

We are interested in better understanding what influences the outcome of an *optimal task tracking* problem, i.e. given a fixed task sequence how do we train a model to optimize the performance on the final target task. To clarify our further writing, we summarize essential elements involved in sequential learning and visualize the overall setting in Fig. 1.

A task $\tau$ consists of a probability distribution $p(x, y)$ of data points and desired outputs as well as a loss function $\mathcal{L} : (p, f) \mapsto \mathbb{R}_+$ that maps an input-output behavior of a model to a cost score. The learning mapping $\omega : (f, \tau) \mapsto f_{new}$, e.g. stochastic gradient descent, uses the information in task $\tau$ and initial model $f_{init}$ to train a model $f_{learn}$ optimized for the task until convergence. The tasks and learning mappings $(\tau_i, \omega_i) \in \mathcal{T}$ are applied sequentially starting from an initial model $f_{init}$ to produce the final target model $f_{target}$ to be evaluated on the target task $\tau_{target}$.

# 4 Plasticity and frozen weights

With the basic building blocks in place, we ask in which cases, even with an infinite number of data points, previous tasks can hinder learning on new tasks. We are interested in understanding what causes a model to be more difficult to adapt to a new task than another. This is evidently the case when some of the model weights $\theta$ cannot be updated when training on the new task even when this would yield a better model. Based on this reasoning, we introduce the concept of frozen weights, i.e. weights that are unchanged when training on a learning task s.t. *plasticity* is clearly lacking.

**Definition 1** (Frozen weights). *We define a* frozen weight $\theta_{frozen}$ *of model $f$ as a parameter that is not changed through learning $\omega$ on task $\tau$. $\theta_{frozen}(f) = \theta_{frozen}(\omega(\tau, f))$*

Associated to frozen weights, *plasticity* captures how well a model can adapt to a new task.

**Definition 2** (Plasticity). *We define* plasticity *as the rate of change of the model parameters subject to gradient descent updates $\mathcal{P} := -\nabla_\theta \mathcal{L}(f(\theta, x), y)$.*

The definitions of frozen weights and plasticity beg the question in which scenarios *frozen weights* can appear. A promising candidate for such a constellation are *mutually frozen weights*.

**Definition 3** (Mutually frozen weights). *For a model $f(\theta; x)$ with parameters $\theta$ and input $x$, we consider $n$ non-overlapping sets of parameters $\{\theta_i\}$ as* mutually frozen *if, for each set, all parameters are zero and if each parameter in each set is exclusively multiplied with all other sets of parameters. We call such parameters n-sparse frozen weights. As the leading example, weights $\theta_i, \theta_j, \theta_k \in \mathbb{R}$ are triple-sparse mutually frozen if, first, the weights $\theta_i, \theta_j, \theta_k$ only appear multiplied together within the model as $\theta_{ijk} := \theta_i \cdot \theta_j \cdot \theta_k$ s.t. $f(\theta; x) = f(\theta^-, \theta_{ijk}; x)$, where $\theta^-$ refers to all parameters except $\theta_i, \theta_j, \theta_k$ and, secondly if $\theta_i = \theta_j = \theta_k = 0$. Given a loss $\mathcal{L}(\theta; x) \to \mathbb{R}^+$ we see that the gradient of $\theta_i$ is proportional to the remaining weights $\theta_j \cdot \theta_k = 0$:*

$$\frac{\partial \mathcal{L}(\theta; x)}{\partial \theta_i} = \frac{\partial \mathcal{L}(\theta; x)}{\partial f(\theta; x)} \frac{\partial f(\theta; x)}{\partial \theta_{ijk}} \frac{\partial \theta_{ijk}}{\partial \theta_i} = \frac{\partial \mathcal{L}(\theta; x)}{\partial f(\theta; x)} \frac{\partial f(\theta; x)}{\partial \theta_{ijk}} \theta_j \theta_k$$

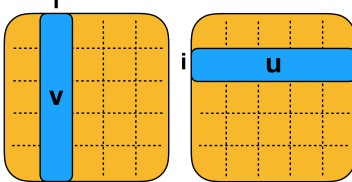 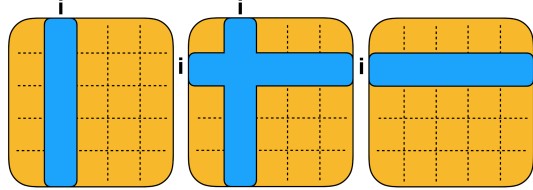

Figure 2: **Scenario for double- and triple-sparse frozen weights: (Left)** Double-sparse frozen weights can occur if, with same index $i$, a zero row vector "u" (sparsity denoted in blue) or zero convolutional filter meets a zero column vector "v" or zero-entries for the associated channel in the following filters. **(Right)** Triple-sparse frozen weights can occur if additionally a diagonal matrix of the same index $i$ entry is zero. Scalar multiplication such as commonly used in normalization layers can serve the part of the diagonal matrix in this case. We are not currently aware of higher-order frozen weight scenarios beyond triple-sparse frozen weights in available NN architectures.

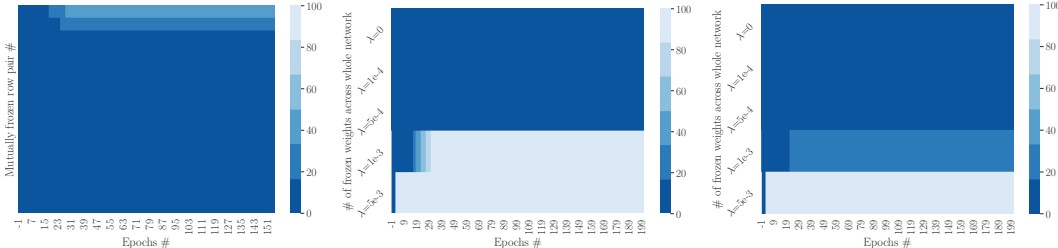

Figure 3: **Existence and dynamics of mutually frozen weights: (Left)** Number of frozen weight filter-pairs between consecutive layers as function of the training epoch in a ResNet18 trained on CIFAR-10 images with weight decay $\lambda =$1e-3. The top rows indicate that, soon after the beginning of training, mutually frozen weights are created in the initial layers (lighter color). **(Middle)** Whether mutually frozen weights are created depends on the amount of weight decay. The heatmap shows the overall number of frozen weights across the entire network for different values of weight decay in a ResNet18 trained on CIFAR-10: only higher values of weight decay lead to the emergence of frozen pairs. **(Right)** Same as center, but for an architecture without skip connections (All-CNN [45]).

*Therefore with condition $\theta_i = \theta_j = \theta_k = 0$ and assuming that the partial gradients $\frac{\mathcal{L}(\theta;x)}{\partial f(\theta;x)}$ and $\frac{\partial f(\theta;x)}{\partial \theta_{ijk}}$ are bounded, the gradient of the mutually frozen weights $\theta_i, \theta_j, \theta_k$ weights are zero.*

We note that *mutually frozen weights* describe n-tuples ($n > 1$) that "freeze together". As depicted in Fig. 2, we will focus on *double-sparse* and *triple-sparse* frozen weights in the remainder of the text. Of course also higher-order freezing involving n-tuples are possible yet do not necessarily contribute further insights. Likewise we deem this less likely to occur in neural network architectures since the influence of individual parameters spreads much further when involving more than three layers.

**Example 1** (Linear regression). *Linear regression does not have "mutually frozen weights". Given $f(\theta;x) = \theta^T x + b, y, b \in \mathbb{R}, x, \theta \in \mathbb{R}^n$, we have $\frac{\partial f(\theta;x)}{\partial \theta} = x$. Hence, unsurprisingly, no weights are multiplied with other weights in a way that could create* mutually frozen weights. *Thus we would not expect linear regression to incur a loss in plasticity in a sequential task learning setup.*

**Example 2** (Neural networks). *A simple example of triple-sparse mutually frozen weights appears for the model $f(\theta_i, \theta_j, \theta_k; x_i) = \theta_i\theta_j\theta_k x$ with $x, \theta_i, \theta_j, \theta_k \in \mathbb{R}$. In this case, one can readily see that if $\theta_i = \theta_j = \theta_k = 0$ then the gradients for all weights will also be zero. This can be regarded as a sub-part of a larger piece-wise linear neural network which will not be further trained due to mutually frozen weights. Note that with 3 or more weights multiplied as shown, the zero gradient is locally stable (see Sec. 4.1).*

The *mutually frozen weights* condition is particularly relevant for deep neural networks which consist of many pairwise multiplications of weights. In practice paired weights need not be zero exactly but simply have very small absolute values compared to the remaining weights. Other situations of training problems due to insufficient gradients exist such as the *vanishing gradient problem* [42], wherein the magnitude of the gradient is reduced as it progresses through the network due to small weight norms. This problem however appears to be largely solved by methods such as identity connections such as in residual networks [43] and normalizations such as batch normalization [44].

From afar it might appear that the answer to the frozen weights problem is simply to avoid them, i.e. to ensure through training a model to not end up with frozen weights. However as we will document in the following, weight decay, which can cause frozen weights, is often connected to better generalization and thus makes frozen weights non-trivial to avoid. This naturally causes a curious trade-off between how we first train a model on a pre-training and subsequently on a target task.

## 4.1 Existence and weight decay dynamics associated with mutually frozen weights

An important question at this point is whether and how such *mutually frozen* weights can occur. In Fig. 3, we show that this indeed can happen in standard training settings (we use the same settings as Mehta et al. [37]). Moreover, we notice that the number of mutually frozen weights *increases* as the training progresses and that the amount of mutually frozen weights created depends on the

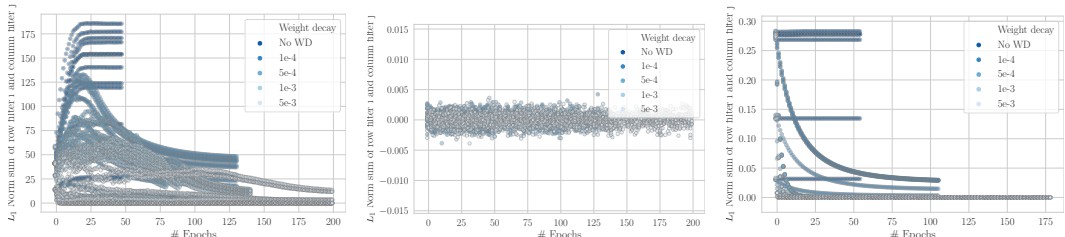

Figure 4: **Dynamics of regular vs mutually frozen weights:** For pairs of column and row filters of an All-CNN trained from scratch on CIFAR-10 we plot the sum of their associated row and column $L1$-norm. **(Left) Reference dynamics.** We select a randomly initialized pair of rows and column weights from each layer and plot the dynamics of the sum of their norms of the weights during training, for different values of weight decay. Larger weight decay is associated with lower overall norms (as expected) but in most cases the weights reach a stable equilibrium aside from zero. **(Middle) Exactly zero dynamics.** We select a specific pair of rows and column weights which are initialized to be exactly zero. We find that these double zero weights remain at zero, that is the weights remain *frozen at zero*. **(Right) Close to zero dynamics.** If the weights are very close but not exactly zero, weight decay is likely to move them back to zero (stable equilibrium).

amount of weight decay applied. While weight decay is not usually expected to create sparsity in the weights for a shallow model [46], we show that for *deep models* weight decay promotes the existence of mutually-frozen weights by modifying the learning dynamics in such a way that makes mutually-frozen weights attractor-points of the dynamics.

For the following argument we pick a triplet of weights $\theta_i, \theta_j, \theta_k \in \mathbb{R}$ which are multiplied for all forward passes of a neural network. Furthermore we assume that the network is piece-wise linear as in the case of ReLU activations. Then we can rewrite the model as $f(\theta; x) = A(\theta^-; x)\theta_i\theta_j\theta_k + B(\theta^-; x)$, where $\theta^-$ indicates that $\theta$ does not contain $\theta_i, \theta_j, \theta_k$. With weight decay, weights $\theta_i, \theta_j, \theta_k$ evolve as follows:

$$\dot{\theta}_{i/j/k} = -\eta\left(\frac{\partial \mathcal{L}(\theta; x)}{\partial f(\theta; x)}\frac{\partial f(\theta; x)}{\partial \theta_{i/j/k}} + \lambda\theta_{i/j/k}\right) = -\eta\left(\frac{\partial \mathcal{L}(\theta; x)}{\partial f(\theta; x)}A(\theta^-; x)\frac{\theta_i\theta_j\theta_k}{\theta_{i/j/k}} + \lambda\theta_{i/j/k}\right)$$

Assuming a quadratic loss[1] $\mathcal{L}(\theta; x) = \frac{1}{2}\|f(\theta; x) - y\|_2^2$, we can further see that $\dot{\theta}_i = -\eta((A(\theta^-; x)\theta_i\theta_j\theta_k + B(\theta^-; x) - y)A(x; \theta^-)\theta_j\theta_k + \lambda\theta_i) = -\eta(A(\theta^-; x)^2\theta_j^2\theta_k^2\theta_i + (B(\theta^-; x) - y)A(\theta^-; x)\theta_j\theta_k + \lambda\theta_i)$. We calculate the stability dynamics of $\theta_i, \theta_j, \theta_k$ linearized around equilibrium weights $(\theta_i, \theta_j, \theta_k)_{\text{eq.}} = (0, 0, 0)$. With this linearization we obtain:

$$\begin{pmatrix} \dot{\theta}_i \\ \dot{\theta}_j \\ \dot{\theta}_k \end{pmatrix} = \eta \begin{pmatrix} -\lambda & 0 & 0 \\ 0 & -\lambda & 0 \\ 0 & 0 & -\lambda \end{pmatrix} \begin{pmatrix} \theta_i \\ \theta_j \\ \theta_k \end{pmatrix}$$

These dynamics have eigenvalues $\alpha_{1,2,3} = -\eta\lambda$. Therefore weights $\theta_i = \theta_j = \theta_k = 0$ are locally stable [47]. The above dynamics calculation was performed on a pair of input-output data $x, y$ but should similarly hold in expectation over the entire dataset. We would expect that *triple-sparse mutually frozen weights* and even higher-order frozen weights once formed remain stable and are not retrained. For double-sparse frozen weights, we find that, under certain conditions, weight can escape the local equilibrium as shown in a similar analysis in Appendix 6.1.

## 4.2 Invariance-plasticity connection

We have established that mutually-frozen weights arise naturally during the training of a DNN, and, perhaps surprisingly, while their effect seems negative as they lock some parameters of the network, their presence is associated with weight decay, which has instead an overall positive regularization effect. To clarify the apparent contradiction, in this section we show that mutually frozen weights may help to codify "learned invariances" to transformations of the data. As invariance we define the sensitivity of a model to local input changes.

---

[1] Note that we can always locally approximate the loss $L(\theta)$ quadratically around the equilibrium point.

**Definition 4** (Invariance - input sensitivity). *Given a smooth input transformation $\varphi_h(x) : x \mapsto \tilde{x}$ applied to input $x$ with $\varphi_{h=0}(x) = x$, we quantify* invariance *as* $\mathtt{Inv} := \frac{\partial f(\theta;\varphi_h(x))}{\partial h}$.

For *composite model functions* $f = f_n(f_{n-1}(\ldots f_1(\varphi_h(x), \theta)$, with parameter vectors $\theta_i$ associated to each intermediate composite function $f_i$, we can find a connection between invariance and plasticity characteristics of a model. For this we first note that via chain-rule, we can decompose the model invariance and model plasticity derivative as follows:

$$\frac{\partial f(\varphi_h(x),\theta)}{\partial h} = \frac{\partial f(\varphi_h(x),\theta)}{\partial f_{n-1}} \cdots \frac{\partial f_{i+1}(\varphi_h(x),\theta)}{\partial f_i} \cdots \frac{\partial f_2(\varphi_h(x),\theta)}{\partial f_1} \frac{\partial f_1(\varphi_h(x),\theta)}{\partial \varphi_h(x)} \frac{\partial \varphi_h(x)}{\partial h}$$

$$\frac{\partial f(\varphi_h(x),\theta)}{\partial \theta_i} = \underbrace{\frac{\partial f(\varphi_h(x),\theta)}{\partial f_{n-1}} \cdots \frac{\partial f_{i+1}(\varphi_h(x),\theta)}{\partial f_i}}_{\text{Shared}} \frac{\partial f_i(\varphi_h(x),\theta)}{\partial \theta_i}$$

From the above we can see that the input sensitivity derivative shares many factors with the model derivative. The earlier the intermediate composite function is applied to the input, the more overlap in factors we have. Thus, for compositional models, input sensitivity and plasticity are naturally linked.

**Lemma 1** (Plasticity-invariance connection). *We assume a composite model $f(\theta, \varphi_h(x)) = f_n(\theta_n, f_{n-1}(\theta_{n-1}, \ldots f_1(\theta_1, \varphi_h(x), \theta)$ and that that the factors unique to the input sensitivity and unique to the model derivative are non-zero $\frac{\partial f_{i-1}(\varphi_h(x),\theta)}{\partial f_1} \frac{\partial f_1(\varphi_h(x),\theta)}{\partial \varphi_h(x)} \frac{\partial \varphi_h(x)}{\partial h} \neq 0$ and $\frac{\partial f_i(\varphi_h(x),\theta)}{\partial \theta_i} \neq 0$. Then a lack of plasticity in some direction $d\theta$ implies that there exists an input transformation s.t. $\frac{\partial f(\varphi_h(x),\theta)}{\partial h} dh = 0$ in which the model is invariant to transformation $\varphi_h$.*

*Proof.* Knowing that the plasticity is zero in $d\theta$ direction and that the individual, non-shared parameter derivative factors are non-zero implies that the shared factors have a non-empty nullspace. Since the shared factors also appear for the model input sensitivity derivative we can make a statement about possible implied invariances. Case 1: The individual factors of the input have a range space that does not overlap with the nullspace of the shared factors. Then we can deduce that the individual factors of the input sensitivity derivative have a non-empty null space. Case 2: The individual input factors have a range space that does overlap with the nullspace of the shared factors. For both cases, there exists a transformation $\varphi_h$ with direction $dh$ for which the model input derivative vanishes $\frac{\partial f}{\partial h} dh = 0$ since there is either a nullspace in the shared or individual factors that projects derivatives to zero. $\square$

The above shows that mutually frozen weights are naturally linked to invariance if the zero parameter derivative arises in shared derivative factors, which is the case for mutually frozen weights. The assumption of shared factors becomes more and more restrictive the later an intermediate composite function is applied such that we would expect that later layers are *easier to retrain* as observed by Achille et al. [8]. In the limiting case of linear regression, there are no shared factors and we would, in a general setting, not expect any relation between input sensitivity and parameter gradients. For very deep networks (especially without skip connections), the relation should be most pronounced for the layers that are closest to the input since more factors are shared.

## 5 Experiments

We explore the ideas and connections of *mutually frozen weights* and *invariance* and their impact in a number of sequential learning settings across different datasets, network architectures, learning rates, and weight decay settings. Detailed specifications will be provided in the appendix. The following subsections are meant to give further evidence for the following statements:

- Mutually frozen weights occur and are different from weights that are zero (sparse) but not mutually frozen.
- Both sufficiently high learning rates and weight decay are essential for the occurrence of mutually frozen weights and final test performance as tested across two different architectures.
- Mutually frozen weights at the beginning of training can be harmful yet can be removed through a "resetting intervention".
- Across a number of task changes, removing frozen weights is beneficial as long as sufficiently many retraining samples are available.
- We provide an analysis summary relating frozen weights, invariance, and performance.

Table 1: **Mutually sparse weights are fundamentally different from sparse weights.** We compare the performance of All-CNNs [45] trained on CIFAR-10 with/without data augmentation of horizontal flip and random shift of magnitude 0.5 while initializing the networks with different types of sparsity. Mono, double-sparse, and triple-sparse weights are randomly set to zero as described in Appendix 6.5 to ensure similar levels of sparsity. Displayed are mean accuracy results with standard deviation across 3 random seeds. **Triple-sparse weights**: We randomly select filter pairs of layers and batch normalization (BN) (triplet) that are directly multiplied together and set all associated weights within those filters to zero. In terms of matrix operations this would be equivalent to setting a row of the first matrix and the associated column of the second matrix as well as its BN weight to zero. **Double-sparse**: For this case we randomly set any weight to zero to arrive at the sparsity probability in the top row. We exempt batch normalization weights from being set to zero since these weights are naturally tied to many other weights thus frequently lead to triple-frozen weights. **Mono-sparse**: For this case we mimic the *double-sparse* case. However to ensure that no weight is multiplied with a paired zero-filter, we add an identity initialization to existing filters - which barely changes the overall sparsity level. It appears that training sparse networks in itself is not difficult, training double and triple-sparse weights is.

| Type / Average sparsity at init. | 0.5 | 0.8 | 0.92 | 0.98 |
|---|---|---|---|---|
| Mono-sparse accuracy | $92.1 \pm 0.2$ | $91.8 \pm 0.4$ | $92.4 \pm 0.1$ | $92.1 \pm 0.0$ |
| Mono-sparse, no data-aug. | $88.6 \pm 0.1$ | $88.6 \pm 0.2$ | $88.5 \pm 0.3$ | $87.1 \pm 2.8$ |
| Double-sparse accuracy | $92.1 \pm 0.1$ | $89.2 \pm 5.2$ | $77.7 \pm 18.4$ | $67.0 \pm 9.0$ |
| Double-sparse, no data-aug. | $88.6 \pm 0.3$ | $85.6 \pm 5.1$ | $74.9 \pm 6.0$ | $63.4 \pm 23.2$ |
| Triple-sparse accuracy | $87.8 \pm 4.9$ | $84.9 \pm 0.5$ | $65.2 \pm 5.6$ | $14.0 \pm 7.0$ |
| Triple-sparse, no data-aug. | $86.8 \pm 0.3$ | $81.9 \pm 0.2$ | $73.1 \pm 2.8$ | $16.0 \pm 10.4$ |

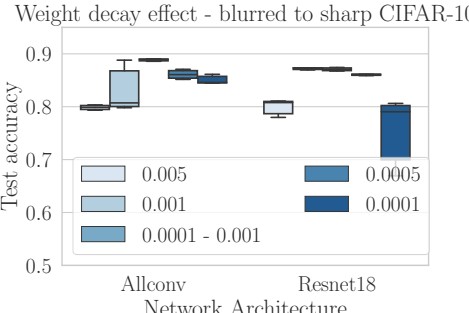 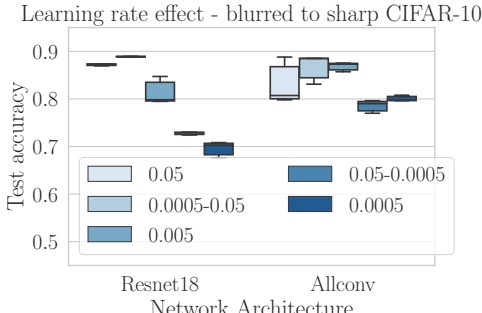

Figure 5: **In a negative pre-training setting, decreasing the pre-training weight decay and pre-training initial learning rate helps since it helps to avoid mutually frozen weights.** For pre-training on blurred CIFAR-10 images and then switching to regular sharp images, we show the test loss for the ResNet-18 and All-CNN architectures across different initial learning rates and weight decay settings. We apply exponential learning rate decay tuned to fit the initial learning rate. **(Left)** Varying weight decay across pre-training while keeping the learning rate schedule constant shows that too much weight decay hurts and too little is similarly detrimental. For negative pre-training it turns out to be better to have little weight decay on the blurred images and more on the standard sharp images. **(Right)** Varying initial learning rates shows that, up to a point, higher initial learning rates yield better retraining results. Even better however is again pre-training with a low initial learning rate and retraining with a higher learning rate.

## 5.1 Mutually frozen weights exist and are different from regular sparse weights

To begin our experiments, we want to demonstrate that mutually frozen weights indeed exist in deep neural network training and that mutually frozen weights are qualitatively different from "regular" sparse zero weights that are not paired with other zero weights. Evci et al. [38] showed that convolutional networks that are sparse can indeed be hard to train to similar test performance as

Table 2: **Retraining performance with/without frozen weights intervention:** We report test accuracies with and without resetting for the baseline case before retraining with mean and standard deviation across 3 seeds as well as the mean number of double/triple frozen weights (FW) (see Appendix 6.2). State-of-the-art results in **bold**. For most tasks with sufficient data, intervening/resetting by removing frozen weights allows pre-trained network to better adapt to a new task. For few samples, we hypothesize that the constraints provided by pre-training are essential when retraining.

| Task change | Model | Reset Test acc. | Test acc. | Double/Triple FW |
|---|---|---|---|---|
| CIFAR-10 [49] → CIFAR-100 [49] | ResNet18 | $74.2 \pm 0.3$ | $73.1 \pm 0.5$ | 60k/50k |
| CIFAR-10 [49] → FashionMNIST | ResNet18 | $96.9 \pm 1.3$ | $95.6 \pm 0.5$ | 100k/90k |
| MNIST [50] → FashionMNIST [51] | ResNet18 | $\mathbf{97.1 \pm 1.0}$ | $96.7 \pm 1.1$ | 200k/170k |
| SVHN [52] → CIFAR-10 [49] | ResNet18 | $88.9 \pm 0.2$ | $87.6 \pm 0.1$ | 3.5M/3.5M |
| ImageNet→FashionMNIST [51] | ResNet50 | $\mathbf{97.8 \pm 1.6}$ | $97.2 \pm 0.2$ | 90k/70k |
| ImageNet→KMNIST [53] | ResNet50 | $\mathbf{99.2 \pm 0.1}$ | $\mathbf{99.2 \pm 0.1}$ | 90k/70k |
| ImageNet→EMNIST (letters) [54] | ResNet50 | $95.6 \pm 0.1$ | $95.3 \pm 0.7$ | 90k/70k |
| ImageNet→CIFAR-100 [49] | ResNet50 | $84.5 \pm 0.2$ | $84.2 \pm 0.3$ | 90k/70k |
| ImageNet→RESISC45 [55] | ResNet50 | $90.2 \pm 0.3$ | $89.7 \pm 0.1$ | 90k/70k |
| ImageNet→Patch Camelyon [56] | ResNet50 | $89.3 \pm 0.6$ | $87.8 \pm 1.4$ | 90k/70k |

networks that are not sparse. However they did not provide a mechanism of how sparsity developed or that *double-sparse* and *triple-sparse* weights in particular make it hard to train.

In Fig. 3 and 4, we count a weight 3-tuple as *mutually frozen* if the absolute value of each weight is smaller than 1e-4. We find that when starting training with a high learning rate and using non-negligible weight decay the number of mutually frozen weights steadily increases with the number of trained epochs as shown in Fig. 3. While weight decay affects all parameters, as shown on the left of Fig. 4, triple-sparse weights remain stably at zero as shown in the middle and right of Fig. 4.

To establish that *mutually frozen weights* are a meaningful concept, we compare the performance when training randomly initialized All-CNNs [45] at different sparsity levels on CIFAR-10 image classification when altering the initial number of small weights. As a point of reference we measured that around $50\%$ of weights fall below our sparsity threshold even without setting any weights to zero. We create *mutually frozen weights* by randomly sampling entire filters, associated filter columns and associated batch normalization weights to be set to zero. "Double-sparse" weights we create by randomly setting weights in a filter to zero but excluding any batch normalization weights. This ensures that triplets of zero weights are very unlikely. Finally, we create "mono-sparse" weights by following the same procedure as for "double-sparse" weights but adding identity filters to the existing filters by "resetting" (subsection 5.2). Thus we can avoid that any weight has an associated filter which is completely zero. As shown in Tab. 1, training a network is hardest with triple-sparse frozen weights, less difficult with double-sparse weights and not difficult with mono-sparse weights. We deduce that, in this setting, sparsity in itself does not necessarily have a strong effect on training performance outcomes, triple-sparse or double-sparse weights however do.

## 5.2   Interventions - Resetting mutually frozen weights - Adding scaled identity

To further test the effect and removal of frozen weights, we devise a simple "resetting" intervention. The intervention adds a small constant times identity initialized weights, similar to the idea of DiracNets [48], to all filters. To the convolutional filters, we add very small zero-mean Gaussian random noise with standard deviation 1e-5 to avoid identical updates by breaking symmetry of updates. Batch normalization (BN) multiplicative weights are either, depending on the magnitude of the intervention, increased by the same small constant across all weights or set directly to 1.0 for all BN weights. By design, this removes all mutually frozen weights since no weight of one filter is associated to only zero weights of an adjacent layer - the effect of which is also shown in Tab. 1.

To demonstrate the effect of the proposed "resetting" intervention, we apply it on a number of pre-training to retraining task changes as recorded in Tab. 2. As long as sufficiently many samples are available for retraining, this can lead to a consistent increase in performance, albeit a small increase if features do not have to be adapted much. For domains that are sufficiently different, we can achieve meaningful gains such as when moving from ImageNet to FashionMNIST, where we surpass the

Table 3: **Re-training performance with and without the use of trainable batch normalization weights:** We report test accuracies with and without using the "affine" option to use trainable multiplicative and additive scalars in batch normalization on CIFAR-10 augmentation transitions. We pick the seminal blurring transition [8] as well as a belated data-augmentation of up to $50\%$ random shift [16] to showcase the influence of frozen weights. In this setting, we compare results of ResNet18, All-CNN, and AlexNet [58] architectures, where we especially note that AlexNet does not have multiplicative normalization weights, e.g. via batch normalization. Intriguingly, without the "affine" option, networks are less prone to negative pre-training yet also perform less well as shown by the performance after "resetting". We calculate the mean number of double- and triple-frozen weights without intervention and before the second task (FW) as described in Appendix 6.2.

| CIFAR-10 Task change | Model | Reset Test acc. | Test acc. | Double/Triple FW |
|---|---|---|---|---|
| Blurred $\rightarrow$ Regular sharp | ResNet18 | $89.0 \pm 0.5$ | $87.2 \pm 0.2$ | 0.4M/0.3M |
| Blurred $\rightarrow$ Regular sharp | No "affine" R18 | $87.4 \pm 0.5$ | $85.1 \pm 3.9$ | 0/0 |
| Blurred $\rightarrow$ Regular sharp | All-CNN | $88.7 \pm 0.2$ | $83.1 \pm 4.9$ | 60k/60k |
| Blurred $\rightarrow$ Regular sharp | No "affine" A-CNN | $85.9 \pm 3.0$ | $88.0 \pm 0.1$ | 0/0 |
| Blurred $\rightarrow$ Regular sharp | AlexNet | $87.3 \pm 0.2$ | $87.5 \pm 0.1$ | 0/0 |
| Regular $\rightarrow$ Random shift | ResNet18 | $94.0 \pm 0.1$ | $93.0 \pm 0.3$ | 90k/80k |
| Regular $\rightarrow$ Random shift | No "affine" R18 | $93.0 \pm 0.0$ | $93.0 \pm 0.1$ | 0/0 |
| Regular $\rightarrow$ Random shift | All-CNN | $90.5 \pm 0.3$ | $89.7 \pm 0.1$ | 4k/4k |
| Regular $\rightarrow$ Random shift | No "affine" A-CNN | $90.5 \pm 0.2$ | $90.4 \pm 0.2$ | 0/0 |
| Regular $\rightarrow$ Random shift | AlexNet | $86.2 \pm 0.6$ | $86.5 \pm 0.5$ | 0/0 |

previous state-of-the-art of $96.9\%$ accuracy [57] and reach $97.8\%$ accuracy. To demonstrate the crucial role that BN weights play, we further provide experiments with and without intervention in Tab. 3 on the "critical learning periods" task of pre-training on blurred images then switching to regular sharp images and the time-sensitive data augmentation task of introducing random shift only after having trained without it. We note that without BN weights, time-sensitive effects are greatly reduced, however the maximal performance is reduced. As a fine-tuning baseline for all experiments, we follow the comprehensive visual benchmark approach [7]. Specific details such as studying the number of samples vs. retraining performance are listed in the appendix.

### 5.3 Learning rate and weight decay - mutually frozen weights

We might wonder whether training could simply be done without weight decay or trained with small learning rates to avoid the formation of many mutually frozen weights. In a *negative pre-training setting* of pre-training on blurred and retraining on sharp CIFAR-10 images [8], we trained either ResNet18 [43] or All-CNNs [45]. With the currently popular setup employing weight decay we find pre-training with low weight decay or initial learning rate but retraining with high initial learning rates and sufficiently high weight decay leads to better test performance as shown in Fig. 5.

### 5.4 Analysis: Invariance-plasticity connection

A compelling part of weight decay is the potential double-edged nature it has which leads to both better test performance yet also to mutually frozen weights. In Fig. 6, we showcase the relationship of performance to the number of frozen weights at the beginning and the end of training. Likewise, we highlight the relation between *invariance* and performance on the very same experiments. To quantify invariance we utilize the Fisher discrimation score [59, 60] and adapt it to analysis of high-dimensional features extracted before the last layer of a Resnet18 where we compute mean $\mu$ and variance $\Sigma$ for each class individually. We are not only interested in absolute invariance but also care about how invariant a model is within a class but also how this variance is scaled relative to the distance between classes.

$$S = \frac{\sigma_{\text{between}}^2}{\sigma_{\text{within}}^2} = \frac{(\vec{w} \cdot (\vec{\mu}_1 - \vec{\mu}_0))^2}{\vec{w}^T(\Sigma_0 + \Sigma_1)\vec{w}} \overset{w=\mu_1-\mu_0}{\approx} \frac{(\vec{\mu}_1 - \vec{\mu}_0)^4}{(\vec{\mu}_1 - \vec{\mu}_0)^T(\Sigma_0 + \Sigma_1)(\vec{\mu}_1 - \vec{\mu}_0)} \tag{1}$$

As shown in Fig. 6, we find that zero initial frozen weights are best to later achieve good performance when retraining. After retraining however, a low but non-zero initial frozen weight count is best while

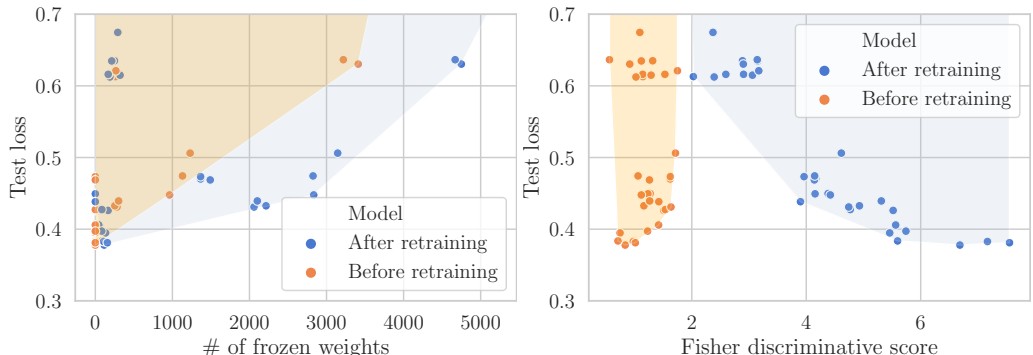

Figure 6: **Plasticity-invariance trade-off:** For the CIFAR-10 dataset, we aggregate results across different weight decay and learning rate settings when pre-training a ResNet18 either on a) blurred or b) on 10% of training to then fine-tuning on the full standard CIFAR-10. We contrast the measured statistics between *before* fine-tuning on CIFAR-10 and *after* fine-tuning on CIFAR-10 but always report the final target standard CIFAR-10 test loss (lower is better). **(Left - performance-frozen weights)** We find that, before fine-tuning, more frozen weights relate to a lower attainable retraining performance. After fine-tuning it appears to be best to have some, but not too many, frozen weights such that a small number of frozen weights leads to the best performance. **(Right - performance-invariance)** We measure the Fisher discriminative score (eq. 1) before and after fine-tuning. Before fine-tuning more invariance, as part of a pareto-front, is associated with lower final performance whereas after fine-tuning a higher fisher score relates to better final performance. The results hint at a possible trade-off between the performance on the pre-training task, which can create a higher initial invariance and create frozen weights, and the ability to retrain to the target task.

more frozen weights decrease performance. When regarding Fisher score *invariance*, the models that are most invariant before retraining appear to incur a performance penalty after retraining, while the models that are more invariant *after retraining* outperform their less invariant counterparts.

# 6 Discussion and conclusion

We started our investigation intrigued by the initial observation that deep networks can exhibit *critical learning periods* [8] such that pre-training on blurred images can lead to worse performance when retraining on sharp images. From this starting point we showed that *mutually frozen weights* play a role in making it harder to retrain neural networks by limiting how gradients flow in a network which is a qualitatively different kind of sparsity that, to our knowledge, has not been discussed before.

Mutually frozen weights may not occur in all commonly used networks. Likewise it is possible that mutually frozen weights can be avoided in the future by improving upon weight decay. Training of a DNN involves numerous other factors which may overshadow the effect of frozen weights and make the analysis more difficult. Due to the potential generality of our proposed concepts it is likewise always helpful to add further datasets and data and model types to assess whether the same effects are present. The scope of this paper is to further the understanding of transfer learning. As such it may have an impact on any setting associated to transfer learning and may help make deep learning solutions more accessible.

Thought-provoking about *mutually frozen weight resetting* is that a very sparse network, whose mutually frozen weights are completely removed, can be trained to the same accuracy as a network trained from scratch. Furthermore, retraining even ImageNet pre-trained models can often be slightly improved by removing existing frozen weights and even helped us obtain state-of-the-art results. Better understanding mutually frozen weights may enable to us to get the most out of pre-trained networks when retraining them on datasets that require a focus on different features. Finally, we touched upon a potentially fundamental trade-off in sequential learning between *invariance* and *plasticity* for *deep* networks. We hope this may spark new research in sequential learning on how best to train a network sequentially, which types of networks are especially conducive to sequential learning and which trade-offs may be inevitable in sequential learning.

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
