# Appendix

To ensure that our results can be replicated we provide further experimental details and complementary results in the following. All experiments where conducted using Pytorch [61] on an internal cluster of up to 8 Titan X GPUs. Hyperparameters were hand-tuned starting with established values such as from the *Visual Task Adaptation Benchmark* [7]. All experiments were run using stochastic gradient descent (SGD) with momentum 0.9. Each experiment was run for 3 different random seeds and we either report each individual result, the mean and variance or a boxplot that captures the changes across random seeds. Experiments were conducted across a span of several months.

Our own code is provided with an MIT license. The principal remaining assets of this work are the used datasets which are commonly used in the community and available for research purposes.

Table 4: Licenses of used datasets and assets

| Dataset or asset | License |
|---|---|
| Own code | MIT license |
| FashionMNIST [51] | MIT license |
| MNIST [50] | Creative Commons Attribution-Share Alike 3.0 |
| EMNIST | Unknown |
| KMNIST | CC BY-SA 4.0 license |
| CIFAR-10 | MIT license |
| CIFAR-100 | Unknown |
| RESISC45 | Fair use exception |
| Patch Camelyon | CC0 License |
| ImageNet-pre-trained Pytorch ResNet models | BSD-style license |

## 6.1 Double-sparse frozen weights stability analysis

We reiterate the same analysis as for triple-sparse weights found in Sec. 4.1 but now focus on double-sparse frozen weights. For this argument we pick a pair of weights $\theta_i, \theta_j \in \mathbb{R}$ which are multiplied for all forward passes of a neural network. Furthermore we assume that the network is piece-wise linear as in the case of ReLU activations. Then we can rewrite the model as $f(\theta; x) = A(\theta^-; x)\theta_i\theta_j + B(\theta^-; x)$, where $\theta^-$ indicates that $\theta$ does not contain $\theta_i, \theta_j$. In other words, weights $\theta_i, \theta_j$ influence some of the model but not all of it. With weight decay weights $\theta_i, \theta_j$ evolve as follows:

$$\dot{\theta}_{i/j} = -\eta\left(\frac{\partial\mathcal{L}(\theta; x)}{\partial f(\theta; x)}\frac{\partial f(\theta; x)}{\partial\theta_{i/j}} + \lambda\theta_{i/j}\right) = -\eta\left(\frac{\partial\mathcal{L}(\theta; x)}{\partial f(\theta; x)}A(\theta^-; x)\frac{\theta_i\theta_j}{\theta_{i/j}} + \lambda\theta_{i/j}\right)$$

Again assuming a mean-squared error (MSE) loss $\mathcal{L}(\theta; x) = \frac{1}{2}||f(\theta; x) - y||_2^2$, we now compute that $\dot{\theta}_i = -\eta((A(\theta^-; x)\theta_i\theta_j + B(\theta^-; x) - y)A(x; \theta^-)\theta_j + \lambda\theta_i) = -\eta(A(\theta^-; x)^2\theta_j^2\theta_i + (B(\theta^-; x) - y)A(\theta^-; x)\theta_j + \lambda\theta_i)$. We now linearize these dynamics of $\theta_i, \theta_j$, which are now qualitatively different from triple-sparse dynamics around equilibrium weights $(\theta_i, \theta_j)_{\text{eq.}} = (0, 0)$. With this linearization we obtain:

$$\begin{pmatrix} \dot{\theta}_i \\ \dot{\theta}_j \end{pmatrix} = \eta\begin{pmatrix} -\lambda & (B(\theta^-; x) - y)A(\theta^-; x) \\ (B(\theta^-; x) - y)A(\theta^-; x) & -\lambda \end{pmatrix}\begin{pmatrix} \theta_i \\ \theta_j \end{pmatrix}$$

These dynamics have eigenvalues $\alpha_{1,2} = \eta(-\lambda \pm |B(\theta^-; x) - y||A(\theta^-; x)|)$. We now have a more involved situation revolving around the local stability of double-sparse weights. Recall that $B(\theta^-; x)$ refers to the part of the network that is unconnected to $\theta_i$ and $\theta_j$. If this part, fits the training labels $y$ well, i.e. $|B(\theta^-; x) - y|$ is small compared to the weight decay factor $\lambda$, then we would expect weights $\theta_i$ and $\theta_j$ to remain close to or at zero. In other words, if there is no need to further fit the training data, then double-sparse weights should remain double-sparse. If however, $|B(\theta^-; x) - y||A(\theta^-; x))|$ is large compared to the weight decay factor, we would expect that re-training of weights is possible

and does occur. This analysis appears to also match our experimental results provided in this work. We conclude that double-sparse weights may be a hindrance, an obstacle, to re-training however not an insurmountable one.

## 6.2 Measuring frozen weights

To measure double or triple-frozen weights within a network, we use the following steps when iterating through the layers of the network:

- We set a "cutoff" at 1e-2 for the $L2$-norm of the row or column in question.
- We count as double-frozen if, for the same index $i$, a row $i$ in the current layer and a column $i$ in the next layer has $L2$-norm lower than the cutoff. We compute the sizes of the row and column and add them to the double-frozen weights count.
- We count as triple-frozen if in addition to the double-frozen condition, we also have a column $i$ in the layer after the previous frozen column layer. In this case, the weights of this column are additionally added to the triple-frozen weight count.

The "cutoff" threshold is chosen conservatively to be close to the mathematical definition of zero norm. Very likely, training issues start to emerge much before the norm decreases below the threshold.

In all experiments, we compute frozen weights after the first training task. In the case of ImageNet pre-training, we report the number of double- and triple-frozen weights of the ImageNet-pre-trained model.

## 6.3 Elements on sequential learning

Complementing section 3, we provide a summary of the *seqential task learning* notation used for *optimal task tracking* in this work.

Table 5: Summary of notation, definitions, and terminology

| Symbol | Definition |
|---|---|
| $x \in \mathcal{X}, u \in \mathcal{U}, y \in \mathcal{Y}$ | Input, model output, and desired output |
| $p(x, y)$ | Joint probability distribution of input x and target or label y |
| $f : \mathcal{X} \times \Theta \to \mathcal{U}, \theta \in \Theta$ | Model function with parameters in parameter space |
| $\mathcal{L}(p, f) : (p, f) \mapsto \mathbb{R}_+$ | Loss objective of accumulated loss of model across distribution |
| $\mathcal{T} = ((\tau_1, \omega_1), ..., (\tau_n, \omega_n))$ | Learning path $\mathcal{T}$ as a sequence of tasks $\tau_i$ and learning processes $\omega_i$ |
| Task $\tau := (p, \mathcal{L})$ | **Definition 5** (Supervised task). *We define a supervised task as the optimization problem associated with the tuple of dataset distribution $p(x, y)$ and loss function $\mathcal{L}(p, f) : (p, f) \mapsto \mathbb{R}_+$, where in this work $\mathcal{L}(p, f) = \mathbb{E}_{p(x,y)}[\ell(f(x), y)]$ with individual data point loss $\ell : \mathcal{U} \times \mathcal{Y} \to \mathbb{R}_+$.* |
| Learn $\omega : (\tau, f) \mapsto f$ | **Definition 6** (Learning). *We define* learning *as a mapping of a task and an initial model $f_{init}$ to a new learned model $f_{learn}$:* $\omega : (\tau, f_{init}) \mapsto f_{learn}$ |
| $\Omega : (\mathcal{T}, f) \mapsto f$ | **Definition 7** (Sequential learning process). *We define a* sequential learning process $\Omega$ *of a* learning path *consisting of a sequence of ordered task-learn process pairs* $\mathcal{T} = ((\tau_1, \omega_1), \ldots, (\tau_n, \omega_n))$ *as the sequential application of the task-learn process pairs to an initial model $f_{init}$.* $f_\Omega = \omega_n(\tau_n, \omega_{n-1}(\tau_{n-1}(\ldots, \ldots, f_{init}) =: \Omega(\mathcal{T}, f_{init})$ |

## 6.4 Sample effect and resetting

As alluded to in the main text, weight resetting to remove mutually frozen weights does not always have a beneficial effect on performance. Removing the restriction only helps when sufficiently

many samples are available and possibly also only when sufficiently different features need to be learned. This trend is illustrated in Fig. 7. In this experiment, we used an ImageNet-pre-trained Resnet50 [43] and then fine-tuned it on the FashionMNIST dataset. The FashionMNIST images were resized from $28 \times 28$ to $224 \times 224$ and transformed to an RGB image by copying the values 3 times (standard colorless RGB image). Additionally, we apply the standard ImageNet normalization (mean $= [0.485, 0.456, 0.406]$, standard dev. $= [0.229, 0.224, 0.225]$). Replicating the fine-tuning approach by Zhai et al. [7], the network is then trained with an initial learning rate of $0.003$ with a decay of $0.97$ per epoch and $0.001$ weight decay. Training was stopped if the validation loss did not improve within the last 20 epochs or if 150 epochs were reached. We can see that for few samples, the standard fine-tuning approach works best. When more samples are available, resetting can improve performance.

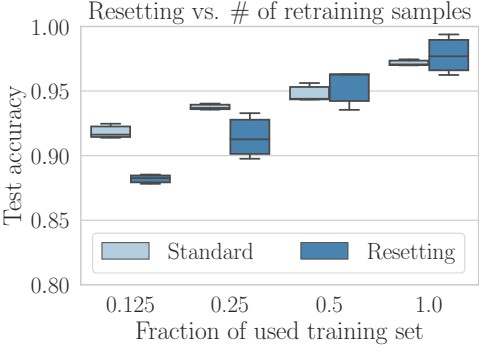

Figure 7: **The value of "resetting" increases with the number of available samples for retraining:** Naturally performance decreases with fewer available samples. However fine-tuning appears to be less robust to fewer labels when an intervention is applied that removes *frozen weights*. We retrain an ImageNet pre-trained Resnet50 on the FashionMNIST [51] dataset with or without an intervention for "resetting" pre-trained weights. For fewer data points, standard fine-tuning without "resetting" leads to better performance. With more available data points, resetting improves performance.

## 6.5   Experimental specifications

Generally, the training set was split into $90\%$ training set and $10\%$ validation set. The test set was kept as is. As mentioned above, hyperparameters were manually chosen, starting at values from the literature, to more quickly reach relevant hyperparameter settings than, for example, random search would.

**Existence and dynamics of mutually frozen weights:** In the first few experiments, we showcase that *mutually frozen weights* exist and plot some of the weight norm dynamics associated to this phenomenon. We ran the associated experiments on CIFAR-10 images classification on the Resnet18 [43] and All-CNN [45] architectures without ImageNet pre-training. No data augmentation is applied. The initial learning rate chosen as $0.05$, with learning rate decay of $0.97$ per epoch. Weight decay was run for all settings in WD $\in \{0.0, 0.0001, 0.0005, 0.001, 0.005\}$. We saved the model parameters after every epoch and then ran an analysis script on the weights to count how many mutually frozen weights were present (Fig. 3). A filter is counted as frozen if the associated batch normalization weights and the associated filter from the next layer are all zero.

**Dynamics figure:** The dynamics figure shows how the norm of the filter "row" plus the norm of the next layer's filter "column" summed together evolve as training progresses. We track selected row-column pair norm sums for all layers of an All-CNN [45] across the various weight decay settings detailed in the previous paragraph. In Fig. 4, we contrast how these weight decay settings affect the norm sum dynamics of filter pairs. As one may expect, one the left, more weight decay leads to smaller weight norms yet converges to non-zero values for most filter pairs, especially for lower weight decay settings. In the middle and right figure however we contrast this picture by studying what happens when we set the weight of filter pairs either directly to zero or to a small non-zero value (standard deviation of zero-mean values $0.0001$). In this case we observe that weight remain at zero or move back to zero norm, respectively.

**Mutually frozen weights vs. doubly frozen vs. mono-sparse weights:** For the experiments detailing that there is a qualitative difference for different types of sparsity we again chose an All-CNN [45] architecture, trained on CIFAR-10 images classification with an initial learning rate of 0.05, exponential learning rate decay of 0.97 per epoch and 0.001 weight decay. To the training data we apply a random shift data augmentation of 50% both horizontally and vertically and a 50%-probability random horizontal flip. Without setting any weights to zero, roughly half of all randomly initialized weights are below the 1e-4 threshold and thus deemed sparse (but not mutually frozen). The results are detailed in Tab. 1.

- **Mutually frozen:** To create mutually frozen weights, we set row-column pairs (and associated batch normalization weights) to zero with probabilities $\{0.2, 0.4, 0.6, 0.8\}$. This leads to overall sparsity levels of $\{0.5, 0.8, 0.92, 0.98\}$, respectively.

- **Doubly sparse:** To arrive at the same sparsity levels, we create a binary mask with which we set weights to zero. As binary-zero probabilities we chose $\{0.5, 0.8, 0.92, 0.98\}$. This again leads to the same overall sparsity levels of $\{0.75, 0.9, 0.96, 0.99\}$.

- **Mono-sparse:** For this sparsity type, we followed the same procedure as for *doubly-sparse* weights yet reintroduced identity mappings to avoid a loss in plasticity. To this end, we add back Dirac-delta function initialized filters to the existing filter weights. To one-dimensional weights such as batch normalization multiplication factors we add the value of 1.0. As the experiments show, this reintroduced the ability to adapt despite high sparsity levels.

**Weight decay and learning rate effects:** For these experiments, we train ResNet18 [43] and All-CNN [45] architectures for various weight decay and learning rate settings. Different to the previous experiments, we first pre-train our model on *blurred* CIFAR-10 images. To create blurred images, we downsample images to $1/4$ of their original size of $32 \times 32$ pixels and then resize back to their original size. We then re-train our model on the standard unblurred CIFAR-10 images. Other than blurring for pre-training and no blurring for re-training, no data augmentation is applied.

- **Weight decay:** We fix the initial learning rate at 0.05 with exponential decay factor 0.95. We run experiments for different weight decay factors WD $\in \{0.0, 0.0001, 0.0005, 0.001, 0.005, 0.0001$ to $0.001\}$, where the last factor denotes that we train with a lower weight decay on blurred images and on a larger one on the sharp standard images later.

- **Learning rates:** We fix weight decay to be 0.001 and choose learning rate and exponential decay factor tuple combinations from $\{(0.05, 0.95), (0.005, 0.95), (0.0005, 0.99)\}$. For some experiments, similar to weight decay, we also change the initial learning rate - exponential decay tuple when pre-training on blurred vs. when re-training on standard target images.

The experimental results to this setup are illustrated in the boxplots in Fig. 5. We observe that both sufficiently high weight decay and learning rates are important. Remarkably however, the best results are obtained when we do not commit on a large initial learning rate or large weight decay on the "corrupted" blurred pre-training task but rather move from a small weight decay or small initial learning to a larger value on the standard unblurred target task.

**Various task changes with and without resetting intervention:** To show that frozen weights exist and are relevant in various deep learning scenarios, we provided further task change scenarios summarized in Tab. 2. In the following, we provide more detailed descriptions of the hyperparameter values chosen for each experiment listed within Tab. 6, Tab. 7, and Tab. 8.[2]

### 6.6 Plasticity-Invariance overview

To create the pareto-front Fig. 6, we collected results from all the learning rate change and weight decay change experiments which are first trained on blurred CIFAR-10 images and then retrained on standard unblurred CIFAR-10 images. Additionally, we include experiments of *warm starting* neural networks training, where we pre-train neural networks with 10% of the total training set and later

---

[2]Patch Camelyon [56] augmentation settings used from `https://github.com/alexmagsam/metastasis-detection`

Table 6: **Retraining performance with and without frozen weights intervention:** We provide more detailed experimental specifications for the task changes and their intervention results in the following.

| Task change | Specifications |
|---|---|
| CIFAR-10: Blurred → Regular sharp | Using a Resnet18 with initial learning rate 0.05, exponential learning rate decay 0.95 per epoch, weight decay 0.001. Training was stopped if the validation loss did not improve for 40 epochs or 200 epochs were reached. Blurring was performed by downscaling images to $1/4$ of their size and upscaling back to the original $32 \times 32$ size. No other data augmentation was used. *Intervention:* We add Dirac-initialized weights multiplied by 0.2 to all weights. Batch normalization weights and other one-dimensional tensors are likewise increased by 0.2. |
| CIFAR-10: Blurred → Regular sharp | Using an All-CNN with initial learning rate 0.05, exponential learning rate decay 0.97 per epoch, weight decay 0.001. All other values as in row above. *Intervention:* We add Dirac-initialized weights multiplied by 0.1 to all weights. Batch normalization weights and other one-dimensional tensors are likewise increased by 0.1. |
| CIFAR-10: Regular → Random shift | Using a ResNet18, we trained first on standard CIFAR-10 images and then later retrained on CIFAR-10 images with random shift data augmentation of 100% of the image size (uniformly sampled horizontal and vertical translation with maximal magnitude of 100% of the image-side length). We train with an initial learning rate of 0.05, learning rate decay 0.95 and weight decay 0.001. Training is stopped after 60 without validation loss improvement or when 200 epochs are reached. *Intervention:* We add Dirac-initialized weights multiplied by 0.4 to all weights. Batch normalization weights and other one-dimensional tensors are set to 1.0. |
| CIFAR-10 [49] → CIFAR-100 [49] | We train a ResNet18 with initial learning rate 0.05, learning rate decay 0.95 and weight decay 0.001. Data augmentation of up to 0.5 horitonzal and vertical shift was used. Training is stopped after 40 without validation loss improvement or when 200 epochs are reached. *Intervention:* We add Dirac-initialized weights multiplied by 0.5 to all weights. Batch normalization weights and other one-dimensional tensors are likewise increased by 0.5. |
| CIFAR-10 [49] → FashionMNIST | We train a ResNet18 with initial learning rate 0.05, learning rate decay 0.95 and weight decay 0.001. No data augmentation is used. Training is stopped after 40 without validation loss improvement or when 200 epochs are reached. *Intervention:* We add Dirac-initialized weights multiplied by 0.2 to all weights. Batch normalization weights and other one-dimensional tensors are increased by 0.2. |
| MNIST [50] → FashionMNIST [51] | We train a ResNet18 with initial learning rate 0.05, learning rate decay 0.95 and weight decay 0.001. No data augmentation is used. Training is stopped after 40 without validation loss improvement or when 200 epochs are reached. *Intervention:* We add Dirac-initialized weights multiplied by 0.4 to all weights. Batch normalization weights and other one-dimensional tensors are likewise set to 1.0. |
| SVHN [52] → CIFAR-10 [49] | We train a ResNet18 with initial learning rate 0.05, learning rate decay 0.95 and weight decay 0.001. No data augmentation is used. Training is stopped after 20/40 without validation loss improvement or when 100/200 epochs are reached, respectively for the SVHN/CIFAR-10 learning step. *Intervention:* We add Dirac-initialized weights multiplied by 0.4 to all weights. Batch normalization weights and other one-dimensional tensors are likewise set to 1.0. |

retrain the networks on 100% of the training set. For these experiments we again vary initial learning rates. All experiments are conducted using the ResNet18 [43] architecture. When varying learning

Table 7: **Retraining performance with and without frozen weights intervention (continued):** We provide more detailed experimental specifications for the task changes and their intervention results in the following.

| Task change | Specifications |
|---|---|
| ImageNet→FashionMNIST [51] | We train a ResNet50 with initial learning rate 0.003, learning rate decay 0.97 and weight decay 0.001. We use the standard ImageNet normalization with mean= $[0.485, 0.456, 0.406]$ and standard dev.= $[0.229, 0.224, 0.225]$ and resize image to size $224 \times 224$. Additionally, we convert grayscale FashionMNIST images to RGB-format by replicating the values three times to form the color-channels. Training is stopped after 20 without validation loss improvement or when 150 epochs are reached. *Intervention:* We now train with 0.05, learning rate decay 0.93 and weight decay 0.001. Before re-training, we add Dirac-initialized weights multiplied by 0.05 to all weights. Batch normalization weights and other one-dimensional tensors are set to 1.0. |
| ImageNet→KMNIST [53] | We train a ResNet50 with initial learning rate 0.003, learning rate decay 0.97 and weight decay 0.001. We use the standard ImageNet normalization with mean= $[0.485, 0.456, 0.406]$ and standard dev.= $[0.229, 0.224, 0.225]$ and resize image to size $224 \times 224$. Additionally, we convert grayscale KMNIST images to RGB-format by replicating the values three times to form the color-channels. We apply data augmentation with random shift with magnitude 0.4, uniformly random shearing with maximal angle $40°$, random scaling of magnitude 0.4, random blurring by downsizing down to $30\%$ of the original size, random rotations of up to $10°$, and random contrast changes with maximal magnitude 0.4. Training is stopped after 25 without validation loss improvement or when 150 epochs are reached. *Intervention:* We now train with 0.003, learning rate decay 0.98 and weight decay 0.001. Before re-training, we add Dirac-initialized weights multiplied by 0.01 to all weights. Batch normalization weights and other one-dimensional tensors are increased by 0.01. |
| ImageNet→EMNIST (letters) [54] | We train a ResNet50 with initial learning rate 0.003, learning rate decay 0.97 and weight decay 0.001. We use the standard ImageNet normalization with mean= $[0.485, 0.456, 0.406]$ and standard dev.= $[0.229, 0.224, 0.225]$ and resize image to size $224 \times 224$. Additionally, we convert grayscale EMNIST images to RGB-format by replicating the values three times to form the color-channels. We apply data augmentation with random shift with magnitude 0.4, uniformly random shearing with maximal angle $40°$, random scaling of magnitude 0.4, random blurring by downsizing down to $30\%$ of the original size, random rotations of up to $10°$, and random contrast changes with maximal magnitude 0.4. Training is stopped after 10 without validation loss improvement or when 150 epochs are reached. *Intervention:* We now train with 0.003, learning rate decay 0.97 and weight decay 0.001. Before re-training, we add Dirac-initialized weights multiplied by 0.05 to all weights. Batch normalization weights and other one-dimensional tensors are increased by 0.05. |
| ImageNet→CIFAR-100 [49] | We train a ResNet50 with initial learning rate 0.003, learning rate decay 0.97 and weight decay 0.001. We use the standard ImageNet normalization with mean= $[0.485, 0.456, 0.406]$ and standard dev.= $[0.229, 0.224, 0.225]$. Training is stopped after 20 without validation loss improvement or when 150 epochs are reached. *Intervention:* We now train with 0.01, learning rate decay 0.95 and weight decay 0.001. Before re-training, we add Dirac-initialized weights multiplied by 0.05 to all weights. Batch normalization weights and other one-dimensional tensors are likewise increased by 0.05. |

Table 8: **Retraining performance with and without frozen weights intervention (continued 2):** We provide more detailed experimental specifications for the task changes and their intervention results in the following.

| Task change | Specifications |
|---|---|
| ImageNet→RESISC45 [55] | We train a ResNet50 with initial learning rate 0.003, learning rate decay 0.97 and weight decay 0.001. We use the standard ImageNet normalization with mean= $[0.485, 0.456, 0.406]$ and standard dev.= $[0.229, 0.224, 0.225]$ and resize image to size $224 \times 224$. No other data augmentation is used. Training is stopped after 20 without validation loss improvement or when 150 epochs are reached. *Intervention:* We now train with 0.003, learning rate decay 0.98 and weight decay 0.001. Before re-training, we add Dirac-initialized weights multiplied by 0.01 to all weights. Batch normalization weights and other one-dimensional tensors are likewise increased by 0.01. |
| ImageNet→Patch Camelyon [56] | We train a ResNet50 with initial learning rate 0.003, learning rate decay 0.97 and weight decay 0.001. We use the standard ImageNet normalization with mean= $[0.485, 0.456, 0.406]$ and standard dev.= $[0.229, 0.224, 0.225]$ and the image size of $96 \times 96$ is kept. Additionally, we convert grayscale Patch Camelyon images to RGB-format by replicating the values three times to form the color-channels. We use random color jitter data augmentation of 0.5 brightness, 0.25 saturation, 0.1 hue, and 0.5 contrast. Additionally, we use random horizontal and vertical flips with probability $50\%$ and a random affine transformation using $10°$ skewing and $5\%$ translation horizontally and vertically. Training is stopped after 20 without validation loss improvement or when 150 epochs are reached. *Intervention:* We now train with 0.05, learning rate decay 0.93 and weight decay 0.001. Before re-training, we add Dirac-initialized weights multiplied by 0.01 to all weights. Batch normalization weights and other one-dimensional tensors are likewise increased by 0.01. |

rates, weight decay is kept at 0.001. When varying weight decay, initial learning rates are 0.05 with exponential decay of 0.95 per epoch.

In the figure, we record the final test loss of the cross-entropy error loss used for training. On the x-axis we provide the number of frozen filters (left) or the Fisher discriminative score (right). The Fisher score is slightly adapted to use $w = \mu_2 - \mu_1$. Usually the value $w$ is rescaled with an expression of the inverses of variances of distribution 1 and 2. Given the high-dimensional features extracted from the ResNet18's penultimate layer we opted to use a simplified version of $w$ to remain computationally feasible.

### 6.7 Previous state-of-the-art on FashionMNIST, KMNIST

While the focus of this work was not to achieve state-of-the-art results, we were able to show that resetting *mutually frozen weights* can improve results in a non-trivial way. We achieve state-of-the-art results on FashionMNIST and KMNIST. We note however that the FashionMNIST state-of-the-art appears to be linked to resetting while the KMNIST state-of-the-art benefits more from ImageNet pre-training and data augmentation. We did not extensively tune hyperparameters and model selection to improve results and would expect further improvements given more time and resources alloted to improving upon the state of the art. The main motivation was to show that resetting weights can lead to meaningful positive changes in results, that the concept of *mutually frozen weights* is also practically relevant.

The previous state-of-the-art in FashionMNIST [57] and KMNIST [62] were achieved using newly proposed neural network architectures. Our approach on the other hand relied on standard established ResNet50 models with ImageNet pre-training and frozen weight resetting. It appears reasonable to assume that these improvements are complimentary and that further improvements can be made when combining these approaches.