# OpenReview forum: "On Plasticity, Invariance, and Mutually Frozen Weights in Sequential Task Learning"
_NeurIPS.cc/2021/Conference — NeurIPS 2021 Poster_

### Official Review · Reviewer_tyuV · 2021-07-12

**Rating:** 4
**Confidence:** 4

**Summary:**

The authors of 'On Plasticity, Invariance, and Mutually Frozen Weights in Sequential Task Learning' introduce 'mutually frozen weights', a new metric for evaluating how well neural networks trained on one task can learn a novel task. They provide preliminary theoretical analysis relating their new metric to weight decay and show how it can be manipulated to facilitate sequential task learning.

**Limitations And Societal Impact:**

The authors provide a theoretical analysis of neural networks and I do not see any negative societal impact from their work.

**Main Review:**

The authors define the phenomena of 'mutually frozen weights' as a triplet of parameters that appear on the neural function only as their product and when they are all equal to zero, they will stay zero for the rest of training. They authors also claim that with the MSE loss and weight decay, all of the parameters being zero is a stable attractor of the training dynamics.

It is very unclear to me how often does it happen when three (or more) parameters only appear as their product. This is a critical point of their work as the 'mutually frozen weights' only happens in that scenario. Interestingly enough, fully connected neural network (i.e. Weights multiplication and bias addition followed by non linearity) do not have this feature, and neither do standard Convolutional neural network composed (they don't even have two parameters only appearing as their product), and even with Batch Norm, I believe we are still left with only two parameters sequentially multiplied. I may be missing something, but still, the authors should provide a clear example when this feature occurs in a common neural architecture rather.

The theoretical analysis in parts 4.1 seems sound aside from the reliance on the MSE loss and weight decay. While I do understand why  all thetai/j/k = 0 is a fixed point in the non weight-decay case, it is not clear why it would occur if it is not an attractor (which it isn't since the eigenvalues of the fixed point would be zero if lambda = 0). An extension to more common loss functions (like, softmax cross entropy) and an explanation of what happens without weight decay (as weight decay is not a very popular regularization measure anymore) would enhance the applicability of this analysis.

The relationship to sequential task learning is also not so clear. In their analysis in 4.2, the authors show that weight gradient and gradients to temporal change in input have common factors and conclude that they are thus interlinked. Due to the familiarity of the community with vanishing gradients (and why we all now use relu instead of tanh and sigmoid) I believe this could simply be stated in a couple of sentences. Devoting an entire section and introducing it as a lemma seems odd, especially considering that the authors definitely lacked space judging from their 2nd and 3rd figures which are hard to read and should be at least twice their current size.

Due to lack of obvious applicability of this work, I recommend it be rejected from Neurips 2021


**Time Spent Reviewing:**

2 hours

---

> ### Author Response · Authors · 2021-08-10
> **Clarifying points raised by reviewer tyuV - Applicability, weight decay, and notation**
>
> The kinds of networks affected by potential frozen weights are very general. As an example consider two fully connected layers: If the row of the first layer and the associated column of the second layer are both zero or close to zero, then gradient flow will be severely restricted - even in this 2-tuple setup.
>
> For a 3-tuple setup, we need the middle layer weight matrix to have a zero row and zero column with the same index as the associated zero row in the first layer and zero column in the third layer. In this case, we again achieve a frozen weight scenario. Having a weight matrix with both zero row and column with the same index is easily achieved when multiplying by diagonal matrices as is the case via the affine weights of normalizations such as batch normalization. Associated with this, we show further below that not using such affine weights for batch normalization reduces the negative effect of sparsity observed in Table 1 as detailed in the response to reviewer 1 (RXdT).
>
> We agree that a sequence of 3-tuples of frozen weights may be less likely to happen without methods such as batch normalization. However, as noted in the answer to reviewer 1 (RXdT), while the theoretical analysis of the stability is easier for 3-tuples, we expect an effect for 2-tuples of frozen weights as well (as also measured in our experiments, see Table 1). We also note that the commonly used batch normalization makes 3-tuples more likely. In fact, the affine transform applied after batch norm multiplies each featuremap independently by a learned scalar. So it suffices for that scalar to be zero to have the same effect as setting all the weights in a row and column of the same index to be zero.
> On this topic, we also refer the reviewer to Mehta et al. [37] for an empirical analysis of the optimization and regularization settings that may induce sparsity during training of a convolutional DNN.
>
> In Table 2, we report the number of 3-tuples of mutually frozen filters in a trained deep network. Even with a conservative cut-off threshold of 1e-4 to consider the norm of a filter “close to zero”, we note that the number is still high (furthermore, we emphasize that this is the number of 3-tuples, each three tuple will contain around 600 weights). Even for pre-trained ImageNet models which are in common usage, there are more than 300 mutually frozen filters that we measured even at the very conservative threshold of counting as zero a filter with norm less than 1e-4. This means that the individual weights in such a filter are much smaller than 1e-4.
> We also note that, as we discuss in the response to reviewer 1 (RXdT), while 3-tuples are necessary to prove stability of the mutually frozen configuration, we do expect an effect, even if less pronounced, for 2-tuples of mutually frozen weights. Indeed, we show this empirically in Table 1.
>
> Stability analysis and weight decay:
>
> Indeed we do not claim that it to be an attractor point when not using weight decay (lambda=0) and we show in the experiments that, as expected from the proposition, weight decay affects the final number of frozen weights. We have updated the text to clarify that the stability results are only valid when using weight decay (lambda > 0). Weight decay is both an integral part of both the standard deep learning setup in a many applications as well as a key component that drives weights towards becoming and staying mutually frozen.
>
> An extension to more common loss functions (like, softmax cross entropy):
>
> Note that for the stability analysis it is enough to study a linear approximation of the dynamics in a neighborhood of the point. Even if the loss function is not the MSE loss, taking a second-order approximation of the loss landscape in a neighborhood of the point would still give us a quadratic loss with rescaled curvature comparable to the MSE we study.
>
>
> While it may be the case that weight decay is not used in some domains, it is still the default regularization technique for resnet architectures in deep learning and it makes a significant difference in the generalization performance of the network. For example, the pre-trained ImageNet models shipped with PyTorch are trained with the following script  https://github.com/pytorch/vision/blob/v0.3.0/references/classification/train.py that uses a relatively large weight decay by default.
>
> "The relationship to sequential task learning is also not so clear. In their analysis in 4.2, the authors show that weight gradient and gradients to temporal change in input have common factors and conclude that they are thus interlinked.
> Due to the familiarity of the community with vanishing gradients (and why we all now use relu instead of tanh and sigmoid) I believe this could simply be stated in a couple of sentences. Devoting an entire section and introducing it as a lemma seems odd, especially considering that the authors definitely lacked space judging from their 2nd and 3rd figures which are hard to read and should be at least twice their current size."
>
> Please note that t at line 158 does not indicate the time, rather it parametrizes the intensity of a transformation of the input, which may be a possible source of confusion. We have updated the text at line 153-154 to clarify that $\phi_t$ parametrizes a family of transformations of the input (e.g., geometric transformations, change of contrast/color, etc) of increasing intensity (with $\phi_0$ = identity).
> More generally, Section 4.2 establishes a relation between frozen weights and invariances by showing that they have a shared factor, i.e. input gradients and weight gradients share common factors in deep networks (but not for example in linear regression). This is a connection between input and parameter sensitivity, and does not directly relate to the vanishing gradient problem. What the relationship aims to establish is that losing plasticity by creating a frozen-tuple may naturally come from the necessity to learn robust invariances to input transformations . While the plasticity-stability trade-off is well established [1], this section proposes that there may be another trade-off between plasticity and invariance for compositional functions such as deep networks.
> [1] Mermillod, Martial et al. “The stability-plasticity dilemma: investigating the continuum from catastrophic forgetting to age-limited learning effects.” Frontiers in psychology vol. 4 504. 5 Aug. 2013, doi:10.3389/fpsyg.2013.00504
>
> Sequential learning, particularly in the form of pre-training and fine-tuning is one of the most common applications of deep learning in computer vision in industrial application. The topic is still widely studied today under the guise of large model pre-training, meta-learning, few-shot learning, and so on. The specific setting that we study (ResNet, weight decay, batch norm, data augmentation, ...) closely reproduce the standard setup used for fine-tuning. Thus, we believe that understanding how to facilitate transfer to new tasks, and giving simple ways to improve the results (such as our weight-resetting method) is indeed an important and highly practical venue of research and will ultimately help foster wider application of deep learning in important domains where less data is available (democratization of deep learning).

---

### Official Review · Reviewer_eMiV · 2021-07-14

**Rating:** 6
**Confidence:** 4

**Summary:**

The paper studies the "optimal task tracking" problem under the training setting where we use the "weight decay" regularization.
More specifically, the paper studies the relationship between plasticity-stability tradeoff and shows that a consequence of training with (large enough) weight decay is the "mutually frozen" weights that can damage the plasticity of the network. Moreover, the paper introduces an intervention to reset the mutually frozen weights and improve the performance.

**Limitations And Societal Impact:**

Yes.

**Main Review:**

### Overall
**Originality: Are the tasks or methods new?** Yes
**Quality: Is the submission technically sound?** Partially
**Clarity: Is the submission clearly written?** Yes
**Significance: Are the results important?** Partially

------
### Strengths
**(1)** In general, I appreciate the works that aim to understand the behavior of neural networks rather than jumping into problem-solving and beating the benchmarks. This paper belongs to this category, and I appreciate the novelty and the approach of the paper.

**(2)** The overall theoretical reasoning seems sound to me.

**(3)** The authors provide the code and details for reproducibility.

-----

### Weaknesses

**(1)** While the introduced concept (mutually frozen weights) seems very interesting, I believe the study could benefit if other contributing factors to this concept were studied. For instance, the authors study the impact of weight decay and (to some degree) learning rate (Sec. 5.3) but since the aim is to study the training dynamics. But, several other common factors may or may not contribute to the impact of mutually frozen weights. While I understand it is infeasible to study every contributing factor, the paper could at very least study the prominent factors in training dynamics such as BatchNorm, Dropout, and Optimizer that is shown to have an impact on training dynamics and stability-plasticity in continual learning [1]. More specifically, [2] shows that L2 regularization with Batch Normalization leads to an Exponential Learning Rate Schedule that depends on the momentum and weight decay rate. According to the code that the authors provided, most of the models use Batch Normalization layers. Hence, I believe an additional study on these factors is needed to improve the paper's insights.

**(2)** In the continual learning literature, weight decay is known to generally hurt the performance regarding the stability and catastrophic forgetting [1,3], which contradicts the paper's claim. While the benchmarks in those studies are different, the problem setting seems similar (i.e., distribution shift). Could the authors discuss the relationship of those findings to the findings of their paper?

**(3)** Regarding Table 4, I have three comments/questions:
   *(3.1)* If I understood correctly, the number of frozen weights is very small compared to the total number of parameters in the network.  How this small number of frozen weights can make such a gap (e.g., ~6.6% on CIFAR-10 -> CIFAR-100)? According to the appendix, the reported numbers are calculated for only one specific learning rate. Could it be because of this?
   *(3.2)* It is difficult to find the real impact of the intervention on performance, given that only one hyper-parameter is used. Could the authors provide the results for at least a few of the rows (e.g., one from each category) where a grid (even small) is used to find the best parameters? The rationale is some reported numbers are lower than the typical performances reported in the other studies. For instance, training a ResNet18 on CIFAR-100 from scratch should give above 70% accuracy. While one can argue that they might use different techniques for improving performance, I suppose given this sub-optimal training setting, it is clear how much real gain the proposed intervention provides.

**(4)** [Misc]: In addition to some grammatical errors (e.g., L38), I believe some figures (2, 3, 4)  are misplaced and very far from where they have been referenced. Also, the y-axis titles of Fig. 4 are not entirely readable. It could improve the readability of the paper if the authors could improve the figures. However, I should note that this did not contribute to my decision and is only a suggestion.

[References]
*[1] Mirzadeh, Seyed Iman, et al. "Understanding the Role of Training Regimes in Continual Learning." Advances in Neural Information Processing Systems, 2020.*
*[2] Li, Zhiyuan, and Sanjeev Arora. "An Exponential Learning Rate Schedule for Deep Learning." Eighth International Conference on Learning Representations, 2020.*
*[3] Lange, Matthias De, et al. Continual Learning: A Comparative Study on How to Defy Forgetting in Classification Tasks. 2019.*


----- Update (After Rebuttal) ----
I believe the authors have addressed most of my questions and comments. Hence, I increase my initial score.

**Time Spent Reviewing:**

3

---

> ### Author Response · Authors · 2021-08-10
> **Addressing points raised by reviewer eMiV - hyperparameter study CIFAR-10 -> CIFAR-100 with data augmentation**
>
> (1)
> We agree that it would be interesting to run ablation experiments, particularly in relation to batch norm, to understand if the effects we measure are due to non-trivial interaction of multiple components. Unfortunately, it is difficult to train modern networks without batchnorm (or similar normalization layers) to a similar high accuracy as with batchnorm. However, in the response to reviewer RXdT we discuss the results of ablating the transform - scalar multiplication and addition - of batch norm (which we expect to have the largest effect on mutually frozen weights). In this experiment, we see that the affine transform indeed appears to be a key component of allowing 3-tuples of mutually frozen weights. Moreover, we show that setting to zero single weights is less damaging than setting to zero tuples of weights under otherwise identical training settings, further strengthening the claim that a large part of the effect is due to mutually frozen weights alone.
>
> (2)
> We believe that our claims actually support the results of [1,3]. Indeed in continual learning (which is a different setup than us, since we do not care about retaining the performance on old tasks), the network needs to allocate new capacity over time to the new tasks. As we show (Fig. 2 and 4 and analysis), training with high weight decay fosters the appearance of mutually frozen weights, which cannot be reallocated for new tasks, effectively diminishing the capacity of the network over time. This forces the network to reuse the remaining (non-frozen) weights for the new tasks, thus increasing catastrophic forgetting, in line with the observations of [1,3]. Thus higher weight decay should be related to higher catastrophic forgetting.
> Overall, our paper also supports the view that higher weight decay will damage the network capacity to allocate resources to new tasks, and hence foster catastrophic forgetting in a continual learning setting.
>
> (3)
> Note that the numbers reported in Table 4 are not the number of frozen parameters, but of frozen 3-tuples of filters. Each frozen 3-tuple of filters contain many more individual weights, e.g. with 64 filters in the previous layer and a 3x3 kernel we have 576 = 3*3*64 mutually frozen weights for a single involved frozen filter. In practice we would have at least 2 filters counted together as mutually frozen -> 1152 = 2*576 (with the addition of a mutually frozen batch normalization layer in between with negligible number of parameters).
> There is also an arbitrary cut-off of 10^-4 for the norm of an entire filter to consider the filter “close-to-zero”. However, the filters could be frozen in practice even if their value is larger (the exact threshold would depend on several factors such as relative magnitude of weights in the network, normalization layer effects via batch normalization, gradient magnitude, weight decay magnitude and likely others).
> Therefore we would expect the number of mutually frozen weights to make up a non-negligible part of the overall parameters.
>
> The lower performance on CIFAR-100 is because we did not use data augmentation for that experiment. Following the suggestion, we have repeated the CIFAR-10 -> CIFAR-100 experiments with standard data augmentation and variations in hyperparameters. Training CIFAR-10 -> CIFAR-100 in this setting gives a higher accuracy of around 73%. However, as before, we observe that resetting mutually frozen weights increases the transfer performance to around 74% accuracy. This confirms that the beneficial effect of resetting the frozen weights is not an artifact of the training procedure.
> Moreover, following the reviewer’s suggestion, we have run the same experiment for multiple combinations of hyperparameters by varying the weight decay and initial learning and choosing whether or not to reset weights after pre-training on CIFAR-10.
>
>
> | Initial Lr | WD | Resetting intervention | Test accuracy mean \pm std |
>
> | 0.05 | 0.001 | No | 73.1 \pm 0.5|  - Reference settings
>
> | 0.05 | 0.001 | Yes | 74.2 \pm 0.3 | - Reference settings
>
> | 0.05 | 0.005 | No | 47.3 \pm 0.5 | - Higher weight decay settings
>
> | 0.05 | 0.005 | Yes | 64.5 \pm 0.5 | - Higher weight decay settings
>
> | 0.05 | 0.0001 | No | 71.9 \pm 1.3 | - Lower weight decay settings
>
> | 0.05 | 0.0001 | Yes | 72.3 \pm 0.5 | - Lower weight decay settings
>
> | 0.2 | 0.001 | No | 57.7 \pm 0.5 | - Larger initial learning rate settings
>
> | 0.2 | 0.001 | Yes | 69.0 \pm 0.5 | - Larger initial learning rate settings
>
> | 0.005 | 0.001 | No | 73.3 \pm 0.5 | - Lower initial learning rate settings
>
> | 0.005 | 0.001 | Yes | 73.3 \pm 0.5 | - Lower initial learning rate settings
>
>
> Again across these hyperparameter studies, which we will include, we see the expected trend explored via Fig. 4 that a large learning rate in pre-training or a large weight decay in pre-training can hurt re-training performance. Applying the resetting intervention allows us to remove deficits in the case of high learning rate and high weight decay and even improve performance upon the standard setting in the reference case (first two rows). For settings with low weight decay and low learning rate no large differences between the standard setting and using reference is observed. This is consistent with our experience of when mutually frozen weights occur and when they can hinder re-training.
> Please refer to [37] for a more extensive study of how the different hyper-parameters affect the sparsity.

---

> > ### Comment · Reviewer_eMiV · 2021-09-03
> > **Response to authors**
> >
> > Thank you for the detailed response. While I think (1) has not been fully addressed, I think the rest of the comments/questions are resolved. Hence, I will increase the score for the paper.

---

### Official Review · Reviewer_RXdT · 2021-07-14

**Rating:** 6
**Confidence:** 3

**Summary:**

In this paper, the authors present a mechanistic phenomenon linked to plasticity problems incurred in sequential task learning. They call this phenomenon "mutually frozen weights", describing a situation where triples (or higher order tuples) of weights that multiply each other in a system, all end up with zero value, which prevents error gradients from influencing their value when learning new tasks. They then argue that such a situation arises in real world setting, and explain how weight decay can promote such a situation. They also demonstrate that such frozen weights play a role in input invariance, and thus should be considered in a tradeoff perspective. Finally, the present a "resetting" method that helps mitigate the problems brought by mutually frozen weights, and perform some numerical experiments to demonstrate the efficacy of their method.

**Limitations And Societal Impact:**

I did not notice any discussion of potential societal impact.

**Main Review:**

Originality: To my knowledge, the concept surrounding mutually frozen weights, and their link to learning trajectory deficiencies in sequential task learning, is novel. The concept is sound and the identification of its impact is new.

Quality & Clarity: The paper is thorough and takes the necessary steps to formally define mutually frozen weights, and test for their impact on training. Experiments are well-documented and follow quality guidelines. However, a number of questions remain unanswered with respect to how extended mutually frozen weights might arise.

1- It is not clear to me why the concept of mutually frozen weights needs 3-tuples of weights to be zero, and not 2-tuples. In Definition 3, couldn't the multple \theta_j*\theta_k be replace by a single zero weight ? Maybe I am missing something here, but it seems a 2-tuple is enough to get the phenomenon. In any case, the reason behind why 3-tuples are needed is not explicitly explained, which would greatly improve clarity.

2- In Definition 3, it is stated that mutually frozen weights are 3-tuples that "only appear multiplied together" in the network. In practice, this seems like a very special situation in deep networks, such as in filters, etc. (as treated in experiments). Indeed, typical layer weights will multiply several other weights and thus, gradients will be able to reach them in principle. Is the phenomenon of mutually frozen weight restricted to this situation? (maybe I am missing something here) Also, is the mechanism of hindered sequential task learning observed in the literature restricted to architectures that allow mutually frozen weights? If not, a discussion about the limitation of  mutually frozen weights should be present.

3- Table 1 describes an experiment with frozen weights AND perturbation meant to test invariance (horizontal flip and random shift of magnitude). Why combine both these effects ? What happens to training with different sparsity scenarios when no perturbations are introduced ?

4- Ablation experiments that measure gradients to mutually frozen weights before and after resetting would strengthen the conclusion that the phenomenon, and the solution, are mechanistically related to improvements.

Significance:
If clarity issues described above are resolved, I believe this contribution is significant in understanding mechanisms that influence training trajectories for sequential task learning.

**Time Spent Reviewing:**

1.5hrs

---

> ### Author Response · Authors · 2021-08-10
> **Clarifying points raised by reviewer RXdT - ablation study without data augmentation**
>
> 1 -
> 2-tuples would indeed suffice to have frozen weights if both weights are exactly zero. However, this situation may not be stable, meaning that if the weights are close to zero but not exactly zero, they may, in principle, grow and become unfrozen. We show analytically that  3-tuples with weight decay are always stable on page 4 of the submission, so that if the weights are close to zero they will revert back to zero. Supporting this point, in Table 1 we show empirically that 3-tuples appear to have a stronger effect on final learning outcomes than 2-tuples. Even if less pronounced, we do expect that 2-tuples have a significant and relevant effect in practice however as again shown in Table 1.
>
> 2 - Indeed, the 3-tuple condition is stronger than the 2-tuple condition, but can still happen naturally in trained deep networks. We refer to the answer to R3 for more details, as well as a quantitative measure of the phenomenon. We provide a more detailed description of when 2-tuples and 3-tuples arise in the answer to reviewer 3 (tyuV).
> [37] ​​Mehta, Dushyant, Kwang In Kim, and Christian Theobalt. "On implicit filter level sparsity in convolutional neural networks." Proceedings of the IEEE/CVF Conference on Computer Vision and Pattern Recognition. 2019.
>
> 3 -
> The geometric transformations that we induce are used as part of the standard data augmentation training procedure for All-CNN. They are not meant to test the invariance, but only to further regularize the training process and ensure that the final test error is comparable with the SotA for that architecture. We also tested the effect of frozen weights when training without data augmentation and observed a qualitatively similar effect, albeit as expected with a lower highest test accuracy.
> We provide the sparsity type ablation study (now without data augmentation) in the following table as mean plus-minus standard deviation across 3 random seeds following the same setup as in the submission, except now running without data augmentation:
>
> Sparsity level | 0.75 | 0.9 | 0.96 | 0.99 |
>
> Mono |  88.6 \pm 0.1 | 88.6 \pm 0.2 | 88.5 \pm 0.3 | 87.1 \pm 2.8
>
> Double | 88.6 \pm 0.3 | 85.6 \pm 5.1  | 74.9 \pm 6.0  | 63.4 \pm 23.2 |
>
> Mutually frozen | 86.8 \pm 0.3|  81.9 \pm 0.2  | 73.1 \pm 2.8 | 16.0 \pm 10.4 |
>
> Mutually frozen but without affine weight in the batch normalization | 87.2 \pm 0.1 | 85.6 \pm 2.1 | 84.9 \pm 1.6 | 55.1 \pm 39.1 |
>
> 	Again we note the same qualitative trend that more sparsity does not or barely hurts “mono-sparse” initialized AllConv networks. “Double-sparsity” does hurt performance the more sparsity exists. Finally, mutually frozen or triple sparse weights are the most harmful to performance. In comparison to the experiments with data augmentation we note that the upper bound in performance in these experiments is, as expected, lower. Otherwise the qualitative trend and differences between types of sparsity remain.
> Importantly, and addressing a point by reviewer 2 (eMiV), we provide a further experiment of training the same network with introducing mutually frozen weights but using batch normalization without the affine transformation weights. For this setting we would expect it to be substantially less likely that 3-tuples of zero weights form. Indeed as shown in the table row above, the result more closely resembles the results of double-sparse weights. A notable difference is that without the affine transformation, the overall maximum performance of the network appears to be decreased. Again this strengthens our argument for differentiating 2-tuples from 3-tuples of zero weights - both of which are an important impediment to training however.
>
> 4-  "Ablation experiments that measure gradients to mutually frozen weights before and after resetting would strengthen the conclusion that the phenomenon, and the solution, are mechanistically related to improvements."
> We have indeed measured the number of frozen weights before and after resetting. After resetting, by design and after measurement, the number of frozen weights is always zero. The number of frozen weight filters (not weights) after pre-training and before resetting is shown in Table 2 for a variety of settings. In the following we show the mean sparsity (in percent) of gradients before and after intervention of networks that have finished training on the task (CIFAR-10 without data augmentation with AllConvNet):
>
> Mono-sparse experiments above
>
> Sparsity level | 0.75 | 0.9 | 0.96 | 0.99
>
> Gradient sparsity in percent before intervention | 54.4| 42.3 | 26.9 | 31 |
>
> Gradient sparsity in percent after intervention | 14 | 15.8 | 25.4 | 10.5 |
>
> Mean number of mutually frozen filters after training | 48 | 43.3 | 68.3 | 127 |
>
>
> Double-sparse experiments
>
> Sparsity level | 0.75 | 0.9 | 0.96 | 0.99
>
> Gradient sparsity in percent before intervention | 47.4| 51.9 | 48.0  | 50.7 |
>
> Gradient sparsity in percent after intervention | 24.0 | 18.2 | 10.6 | 8.4 |
>
> Mean number of mutually frozen filters after training | 39.7 | 46.7 | 74 | 170 |
>
>
> Mutually frozen weights sparse experiments
>
> Sparsity level | 0.75 | 0.9 | 0.96 | 0.99
>
> Gradient sparsity in percent before intervention | 79.7| 95 | 99.3  | 99.99 |
>
> Gradient sparsity in percent after intervention | 34.7 | 41.6 | 43.9 | 83.2 |
>
> Mean number of mutually frozen filters after training | 1140.3 | 1786.7 | 2122 | 2136 |
>
> From the data we conclude that the intervention drastically reduces gradient sparsity in the model (we count as sparse a gradient of weight below 1e-4). This number of course only reflects the instantaneous gradient right after the resetting intervention of adding Dirac or identity-initialized weights. We would expect that training ability or plasticity is mostly or fully restored in practice upon training the network further.

---

> > ### Comment · Reviewer_RXdT · 2021-08-20
> > **response to rebuttal**
> >
> > I thank the authors for the detailed response. I have some follow up questions:
> >
> > 1. While your explanation of the role of 2-tuples makes sense, I still believe it is not at all clear from the main text, despite the results from Table 1. What changes do you plan to make to the main manuscript to clarify this point?
> >
> > 2. With respect to the applicability of this result, I have read your response to my, and reviewer tyuV's question concerning batch norm. Again, I would like to better understand what changes to the main text you intend to make to accurately communicate the situation. Also, given that the most likely real-world scenario where 3-tuples give rise to mutually frozen weights appear to be ResNets with weight decay, a discussion about why should these models be used for continual learning to begin with, is in order. Indeed, while you point to pretrained ResNets shipped with Pytorch, their use specific to multi-task setting could be better addressed. Again, the goal here is to identify real-world applicability of your result.
> >
> > 3. With respect to the novel results provided in response to my review, what exactly does "Gradient Sparsity" mean ? How is it measured ?

---

> > > ### Author Response · Authors · 2021-08-23
> > > **Proposed changes in response to review comments**
> > >
> > > Thank you for the questions. Following up on our discussion, we would make the following changes to our paper to clarify those points:
> > >
> > > 1)
> > > We will improve the extent and clarify our results concerning 2- and 3-tuples. In particular:
> > >
> > > - We will change Definition 3 and present double-sparse and triple-sparse weights as separate definitions, and further clarify them with examples of how they can be obtained.
> > >
> > > - Measuring and reporting occurrence of these types of weight combinations for a number of different pre-trained computer vision architectures
> > >
> > > - Complementing this distinction between 2-tuple and 3-tuple (2-sparse and 3-sparse) weights, we will extend the stability analysis to talk both about  2-tuple (in which case we can prove it is a fixed point of the dynamic) and 3-tuple (in which case we can prove that it is also a stable point)
> > >
> > > - We will augment existing experimental results with 2-tuple experiments.  This can be achieved by resetting only the batch normalization multiplicative weights. This way a row filter in a first layer and a column of filters in a second layer may still have double-sparse associated filters, however a triple-sparse case will be very unlikely. Additionally we will also provide the same experiments with the “affine option” for batch normalization layers turned off.
> > >
> > >
> > > 2)
> > > We will add a discussion of how different architectural choices affect our results, and further emphasize results on different architecture (AllCNN, ResNet) and different settings (batch norm, weight decay). Regarding ResNet, our results on AllCNN suggest that it is also affected by the Frozen Weights problem (indeed, even more than ResNet) so ResNet is, overall, still a better choice for continual learning due to both its stronger baseline performance and its empirically better resiliency to frozen weights. We also note that frozen weights are not always negative, since they codify learned invariances and may, therefore, improve results in continual learning in some settings. We will provide a discussion of common architectures in continual learning related to 2-tuple and 3-tuple sparsity.
> > >
> > > 3)
> > > A parameter gradient in the provided table is counted as “sparse” when it is smaller in magnitude than 1e-4. The percentage of sparse gradients is then the number of such “sparse gradients” divided by the total number of parameters. This method of measuring is simple and provides an indicator of how “sparse” the gradients of the model before and after are. We tuned the threshold of what we consider sparse (1e-4) in this case and found that, surprisingly, the sparsity percentage did not change much when changing the threshold value (e.g. to 1e-3 or 1e-5). We will clarify this in the paper.

---

> > > > ### Comment · Reviewer_RXdT · 2021-08-25
> > > > **response**
> > > >
> > > > I would like to thank the authors for addressing my concerns. I am satisfied with the responses and will augment my score by one point. The reason why I do not more strongly advocate for this paper is for its limited demonstration of real-world applicability. Although the author plan to address this topic via the addition of discussion points, I do believe a more complete demonstration of the role of mutually frozen weights across often used architectures in sequential learning, would considerably strengthen the contribution.

---

### Decision · Program_Chairs · 2021-09-27

**Decision:**

Accept (Poster)

**Comment:**

This paper studies the learning dynamics in deep networks by making a novel observation regarding weight decay as mutually frozen weights and their role in generalization. The paper initially received reviews that tended towards rejection. The reviewers had difficulty understanding some details and were concerned whether the results will hold true in real-world settings. The authors provided a thoughtful rebuttal that addressed the reviewers' concerns. The paper was discussed and all the reviewers updated their reviews in the post-rebuttal phase. All reviewers switched their score to weak acceptance (note one reviewer still has a score of 4 in the main review, but they have switched to 6 in comments). AC agrees with the reviewers and suggests acceptance. However, the authors are requested to look at reviewers' feedback and incorporate their comments in the camera-ready.